

# Bridging the gap between sustainability and profitability: unveiling the untapped potential of sea cucumber viscera

Muhammad Fatratullah Muhsin[1], Yushinta Fujaya[2], Andi Aliah Hidayani[2], Hanafiah Fazhan[1,3,4,5], Wan Adibah Wan Mahari[1], Su Shiung Lam[1], Alexander Chong Shu-Chien[4,6], Youji Wang[7], Nor Afiqah-Aleng[8], Nita Rukminasari[2] and Khor Waiho[1,3,4,5]

[1] Higher Institution Centre of Excellence (HICoE), Institute of Tropical Aquaculture and Fisheries, Universiti Malaysia Terengganu, Kuala Nerus, Malaysia
[2] Faculty of Marine Sciences and Fishery, Hasanuddin University, Makassar, Indonesia
[3] STU-UMT Joint Shellfish Research Laboratory, Shantou University, Shantou, China
[4] Centre for Chemical Biology, Universiti Sains Malaysia, Penang, Malaysia
[5] Department of Aquaculture, Faculty of Fisheries, Kasetsart University, Bangkok, Thailand
[6] School of Biological Sciences, Universiti Sains Malaysia, Penang, Malaysia
[7] International Research Center for Marine Biosciences, Ministry of Science and Technology, Shanghai Ocean University, Shanghai, China
[8] Institute of Marine Biotechnology (ICAMB), Universiti Malaysia Terengganu, Kuala Nerus, Malaysia

Corresponding authors
Yushinta Fujaya,
yushinta.fujaya@unhas.ac.id
Khor Waiho, waihokhor@gmail.com

## ABSTRACT

Sea cucumbers have high economic value, and in most forms of trade, their body wall is typically the only part that is harvested and sold. The organs of the sea cucumber, collectively known as the viscera, are frequently discarded, contributing to land and water pollution. However, discarded sea cucumber viscera contain various nutrients that can be used in many applications. Therefore, this review highlights the biological and economic aspects of sea cucumbers, followed by a critical discussion of the nutritional value of their internal organs and possible applications, including as functional feed additives in the aquaculture industry, sources of natural testosterone for application in sex reversal and production of monosex population, of neuroprotective agents against central nervous system disorders and of cosmetic ingredients, especially for skin whitening and anti-ageing products. The review further highlights the valorisation potential of viscera to maximize their economic potential, thus providing an enormous prospect for reusing sea cucumber waste, thereby reducing the negative impact of the sea cucumber fishery sector on the environment.

## INTRODUCTION

Sea cucumber is a valuable seafood item, particularly in Asia, where it is not only priced as an exotic and costly food but also valued for its medicinal and tonic properties (*Rahantoknam, 2015*; *Hossain et al., 2022*). Generally, sea cucumbers are soft-bodied,

elongated, worm-shaped echinoderms with a leathery texture and jelly-like body, resembling a cucumber (*Conand, 1990*; *Zhang et al., 2019*). The sea cucumber fishery supports the livelihood of coastal communities in Asian countries, and with rising market demand, the global fishery production of sea cucumbers has reached 100,000 tons annually (*Ardiansyah et al., 2020*). According to the Food and Agriculture Organization (FAO) global statistics reported in 2019, Indonesia is the largest supplier of wild sea cucumbers among all Southeast Asian countries (*Southeast Asian Fisheries Development Center (SEAFDEC), 2022*).

There are a total of 1,200 known species of sea cucumbers scattered across the world's oceans (*Aydin, 2018*). Among them, many species within the genus *Holothuria* are considered valuable (*Samyn, Massin & Vandenspiegel, 2019*). In most Southeast Asian countries, the preferred high-value sea cucumber species are *Holothuria scabra* (sandfish) (Fig. 1A) and *Stichopus horrens* (Fig. 1B) (dragonfish) (*Southeast Asian Fisheries Development Center (SEAFDEC), 2022*). Wild populations of these species are declining due to extensive commercial exploitation in coastal waters, driven by high demand from both domestic and international markets (*Yaghmour & Whittington-Jones, 2018*; *Kamaruddin & Rehan, 2015*). From a nutritional point of view, sea cucumbers are low in sugar, fat, and cholesterol, but rich in proteins and essential amino acids (*Senadheera et al., 2023*). Additionally, sea cucumbers contain various essential nutritional components, including vitamins, minerals, collagen, and polyunsaturated fatty acids (*Liu et al., 2021*). Consequently, they are harvested and used as food and raw materials. Sea cucumbers have been traditionally consumed as tonic food and used to produce tonic medicine, such as 'gamat' oil in Indonesia and Malaysia. The international sea cucumber market primarily focuses on Asia and the Indo-Pacific region (*Elvevoll et al., 2022*).

Coastal communities have been exploiting various sea cucumber to develop processed products, including dried sea cucumber (Fig. 1C), smoke sea cucumber (Fig. 1D), and sea cucumber crackers (Fig. 1E). Unfortunately, the dwindling supply of sea cucumbers and the ever-increasing demand have forced local fishermen to harvest sea cucumbers of all sizes, which poses a threat to their sustainability (*Southeast Asian Fisheries Development Center (SEAFDEC), 2022*). In addition, sea cucumbers play important roles in the ecosystem. Over-exploitation of sea cucumbers can have several adverse effects, such as compromising sediment health, reducing the ecosystem's capacity to recycle nutrients and resist ocean acidification, diminishing the biodiversity of associated symbionts, and impeding the movement of organic matter from higher trophic levels (*Purcell et al., 2016a*; *Pierrat et al., 2022*). Owing to their high market demand, sea cucumber aquaculture is gaining momentum. The main techniques used in sea cucumber culture include pond farming, pen culture, marine ranching, and tank culture. Sea cucumber aquaculture serves as a viable alternative to relieve the strong fishing pressure on wild sea cucumber populations. However, efforts are still needed to develop sustainable sea cucumber aquaculture programs.

In most forms of trade, only the body wall and muscle bands of sea cucumbers are harvested and sold off (*Senadheera et al., 2023*), while the intestines, gonads, and other organs (termed viscera), which can account for up to 50% of the sea cucumber's total

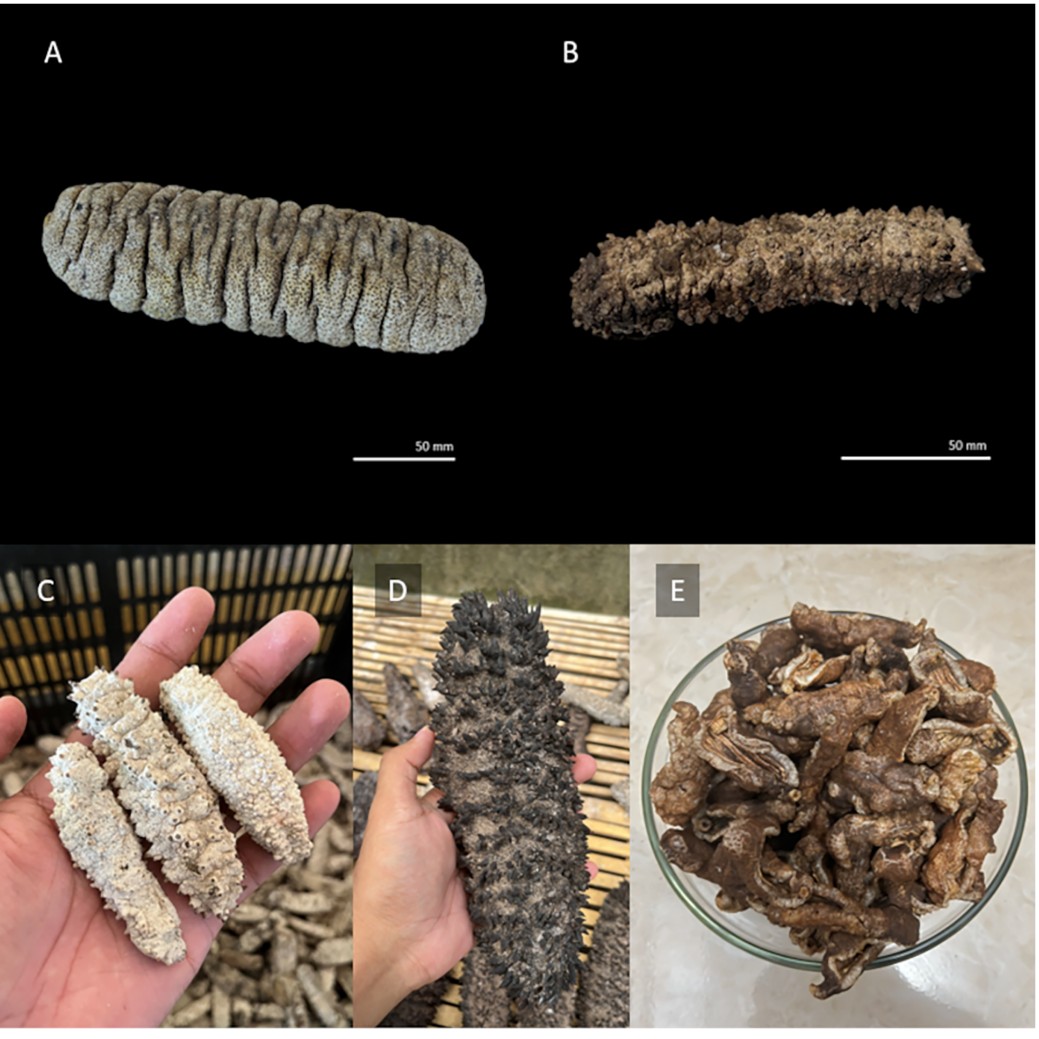

**Figure 1** **Sea cucumbers and their products.** (A) *Holothuria scabra*; (B) *Stichopus horrens*; (C) dried sea cucumber (*Stichopus horrens*); (D) smoke sea cucumber (*Thelenota ananas*); (E) sea cucumber crackers (*Holothuria edulis*).

weight, are considered unwanted products and are often discarded (*Oktaviani, Mulyani & Rochima, 2015*; *Hossain et al., 2022*). The discarding of sea cucumber viscera results in resource waste and environmental contamination since they may contain heavy metals such as arsenic, cadmium, lead, and mercury, which are well-known to be toxic to the environment and human health when exceeding standard limits (*Babji et al., 2020*). Sea cucumber viscera have been reported to contain various nutrients such as oligosaccharides, saponins, phenols, flavonoids, lipids, proteins, fatty acids, and amino acids (*Zhang & Chang, 2014*). They also contain high levels of omega-3 PUFAs and glycine, making them suitable for processing into functional foods, dietary supplements, and pharmaceuticals (*Liu et al., 2021*). Sea cucumber viscera are considered a delicacy and are consumed raw, dried, and fermented in some countries, such as Samoa and Japan (*Eriksson et al., 2007*; *Charan-Dixon et al., 2019*; *Nishanthan et al., 2021*). According to
*Babji et al. (2020)*, consumption of relatively small amounts of sea cucumber viscera hydrolysate may satisfy various vitamin needs in both animal and human nutrition.

As sea cucumber viscera are known to contain various nutrients and bioactive compounds, further research should be conducted to valorise them for the industrial production of high-value nutritional products while addressing the issue of harmful heavy metal content. Even with high heavy metal contents, it is possible to formulate sea cucumber viscera extracts to meet the maximum legally permissible requirements. Additional processing techniques, such as ion-exchange chromatography, can be employed to reduce or eliminate heavy metals during the extraction of sea cucumber viscera (*Ayangbenro & Babalola, 2017*; *Babji et al., 2020*). Therefore, this review summarizes and discusses the importance of sea cucumbers by addressing their general biology, the nutritional content of various body parts, and their socioeconomic contribution. The review further focuses on the often-discarded viscera and explores their nutritional aspects. Additionally, future directions are suggested for recycling viscera, aiming to turn waste into wealth.

## SURVEY METHODOLOGY

Literature searches were conducted on the Web of Science (https://www.webofscicence.com/) and Scopus (https://www.scopus.com) database using the PRISMA method (*Page et al., 2021*) (Fig. 2). Keywords and phrases such as "distribution of sea cucumber", "nutritional value of sea cucumber", "sea cucumber viscera", and "nutritional value of sea cucumber viscera" were used to search for publications in both databases. All the pertinent articles were thoroughly examined after the initial screening to ensure that they were all relevant to the topic.

### Sea cucumber biology

Sea cucumbers belong to the phylum Echinodermata, along with other marine invertebrates that exhibit radial symmetry (*Oh et al., 2017*). They are deposit feeders and play a crucial role in coastal mariculture by directly contributing to the recycling of nutrients and the breakdown of detritus and organic matter (*Gao et al., 2011*; *Zamora et al., 2018*). Sea cucumbers have numerous shield-like buccal tentacles around the mouth, which are enclosed in the external oral hood (Fig. 3). The body wall of sea cucumbers consists of a thick layer of collagenous connective tissue that envelops and protects their internal organs (*Slater & Chen, 2015*). Sea cucumbers are found on practically all substrates, including sand, muddy sand, and sandy mud near seagrass, with depths ranging from 1 to 40 m (*Lewerissa, Uneputty & Waliulu, 2021*; *Liubana, Surbakti & Tubo, 2022*). They are widely distributed in all oceans (Table S1), inhabiting regions from the Arctic to the tropics, and their habitats vary from the intertidal zone to the deepest seas, such as the bottom of the Mariana Trench (*Gonzales-Duran et al., 2021*; *Liu, Xue & Li, 2022*).

Asexual reproduction is possible in sea cucumbers through a process known as fission. They can divide themselves along the median line, where the anterior and posterior ends spin in opposite directions. After a while, the two ends slowly move away from each until they rip the body wall apart, resulting in the division of the organism into two distinct

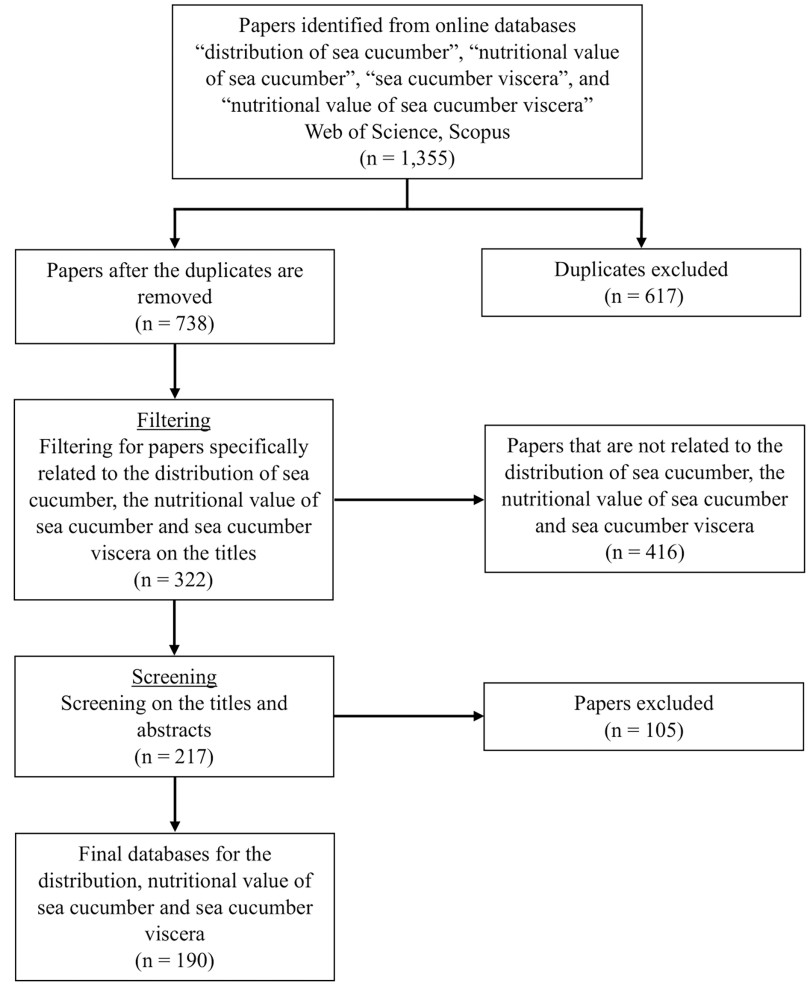

**Figure 2 PRISMA flow diagram of study selection.**

individuals (*Al-Rashdi, Eechaut & Claereboudt, 2012*). Factors such as failure of sexual reproduction, eutrophication, malnutrition, and environmental stimulation, such as drought during prolonged low tides, can all contribute to the occurrence of asexual reproduction in sea cucumbers (*Widianingsih, Hartati & Endrawati, 2014*).

In the wild, sea cucumbers naturally gather in groups consisting of more than ten individuals spaced approximately 5 m apart to perform simultaneous spawning. This synchronized behaviour ensures the highest possible fertilization rates during sexual reproduction (*Rahman & Yusoff, 2017*). Sea cucumber eggs are externally fertilized when the male and female gametes fuse in the water column, as shown in Fig. 4. Fertilized eggs quickly progress to the blastula stage within an hour after fertilization, and by the end of the day, they reach the typical gastrula stage. After two days, the fertilized eggs transform into planktonic auricularia larvae, which exhibit a pelagic habit and feed on suspended microalgae (*Kumara et al., 2013*). After approximately two weeks, sea cucumber larvae will reach the doliolaria stage (non-feeding stage). From there, they metamorphose into the

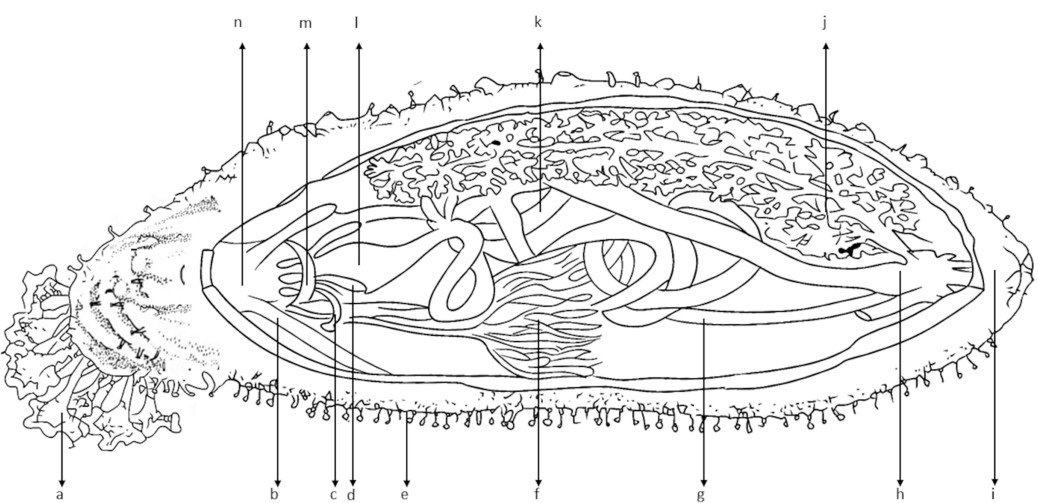

**Figure 3 Anatomy of sea cucumber (*Holothuria scabra*).** Abbreviation: a, tentacles; b, pharynx retractor muscles; c, stone canal; d, polian vesicle; e, tube feet (podia); f, gonad (much enlarged at sexual maturity); g, longitudinal muscle bands; h, cloaca; i, anus; j, respiratory tract; k, intestine; l, stomach; m, ring canal; n, pharynx.

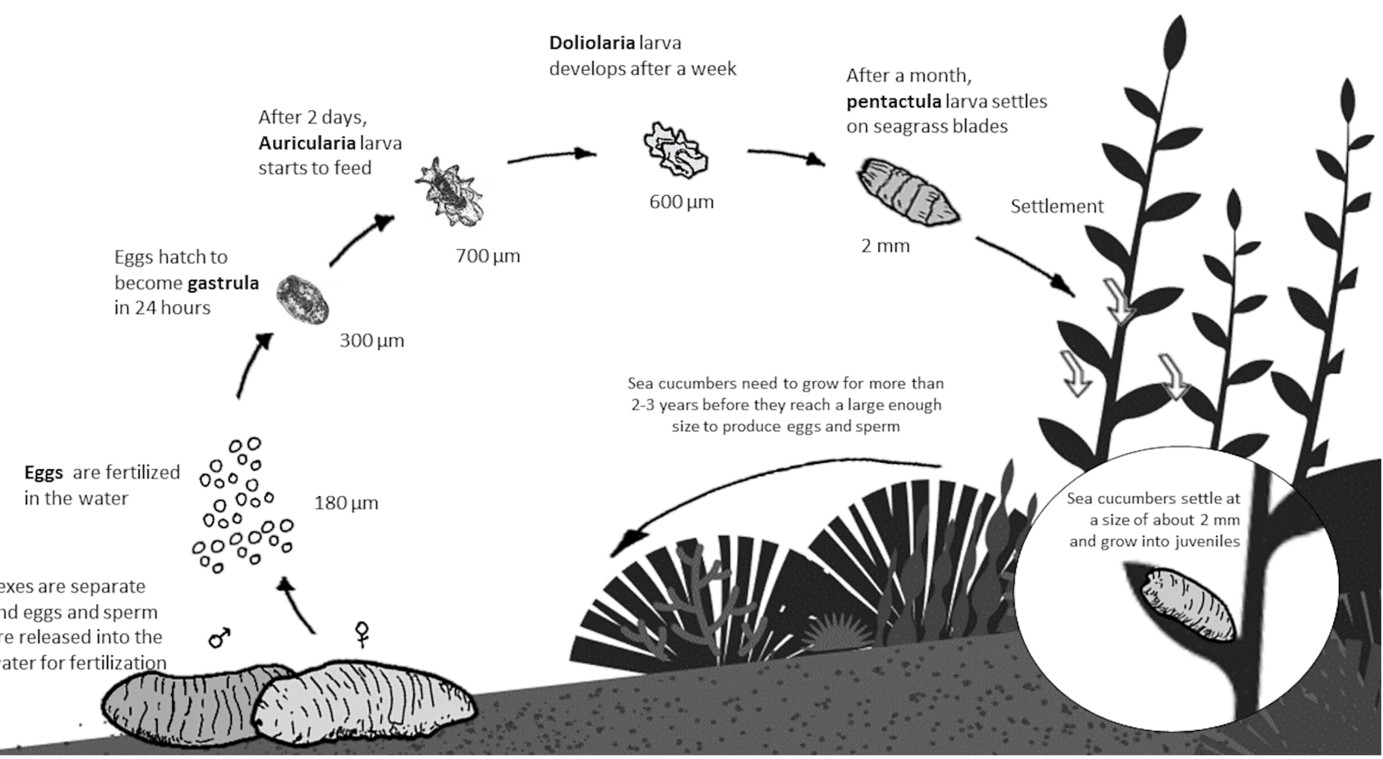

**Figure 4 Lifecycle of sea cucumber (*Holothuria scabra*).**

pentactula stage and subsequently into early juveniles. At this stage, they begin to feed by grazing on the biological film covering the leaf surface. As they grow and mature into adults over another week or two, then move down into the sediment and begin grazing in deeper areas (*Altamirano & Rodriguez, 2022*).

**Table 1 Proximate composition of sea cucumber and sea cucumber viscera (%).**

| Species | Moisture | Protein | Lipid | Ash | Carbohydrates | References |
|---|---|---|---|---|---|---|
| **Sea cucumber** | | | | | | |
| *H. arenicola* (f) | 93.01 | 4.40 | 0.60 | 2.01 | – | *Barzkar, Fariman & Taheri (2017)* |
| *H. atra* (f) | 78.34 | 42.32 | 1.12 | 2.38 | 0.87 | *Oedjoe (2017)* |
| *H. edulis* (f) | 78.16 | 41.61 | 1.08 | 2.47 | 1.14 | *Oedjoe (2017)* |
| *H. impatiens* (f) | 78.41 | 39.94 | 1.12 | 2.16 | 1.37 | *Oedjoe (2017)* |
| *H. lessoni* (d) | 13.47 | 41.18 | 3.02 | 34.51 | 7.86 | *Andriamanamisata & Telesphore (2019)* |
| *H. leucospilota* (f) | 81.24 | 45.71 | 4.60 | 4.30 | 44.96 | *Omran (2013)* |
| *H. mammata* (f) | 85.24 | 7.88 | 0.09 | 5.13 | – | *Aydin et al. (2011)* |
| *H. nobilis* (f) | 76.05 | 42.54 | 1.09 | 2.39 | 0.56 | *Oedjoe (2017)* |
| *H. parva* (f) | 67.92 | 17.61 | 2.43 | 32.74 | – | *Salarzadeh et al. (2012)* |
| *H. sanctori* (f) | – | 8.02 | 0.55 | – | – | *Gocer, Olgunoglu & Olgunoglu (2018)* |
| *H. scabra* (f) | 76.03 | 44.07 | 1.02 | 2.01 | 0.45 | *Oedjoe (2017)* |
| *H. tubulosa* (f) | 80.77 | 7.07 | 10.21 | – | – | *Zmemlia et al. (2020)* |
| *P. australis* (f) | 74.92 | 20.22 | 1.42 | 2.58 | 0.86 | *Widianingsih et al. (2016)* |
| *P. californicus* (d) | 4.03 | 47.03 | 8.19 | 25.73 | 15.02 | *Bechtel, Oliveira & Smiley (2012)* |
| *P. parvimensis* (f) | 90.80 | 4.70 | 0.30 | 3.40 | – | *Chang-Lee, Price & Lampila (1989)* |
| *S. herrmanni* (d) | 8.00 | 67.90 | 1.09 | 17.24 | 7.34 | *Shojaei, Ebrahimi & Nazemi (2020)* |
| *S. horrens* (f) | 92.80 | 3.47 | 0.39 | 3.40 | – | *Barzkar, Fariman & Taheri (2017)* |
| *S. japonicus* (f) | 92.00 | 7.70 | 0.20 | 2.70 | – | *Tanikawa (1955)* |
| *S. variegatus* (d) | 6.27 | 34.33 | 1.08 | 4.34 | – | *Ridhowati et al. (2018)* |
| *S. vastus* (f) | 19.46 | 38.70 | 0.38 | 34.04 | 7.42 | *Rasyid (2017)* |
| **Sea cucumber viscera** | | | | | | |
| *A. japonicus* (f) | 89.54 | 2.20 | 2.12 | 3.94 | – | *Lee et al. (2012)* |
| *C. frondosa* (ad) | 7.84 | 42.20 | 23.68 | 11.51 | – | *Liu et al. (2021)* |
| *C. frondosa* (fd) | 1.79 | 46.12 | 22/77 | 12.31 | – | *Liu et al. (2021)* |
| *C. frondosa* (f) | 82.07 | 8.65 | 4.68 | 2.14 | – | *Liu et al. (2021)* |
| *C. frondosa* (f) | 92.3 | 4.5 | 2.0 | 0.7 | – | *Mamelona, Louis & Pelletier (2010)* |
| *I. japonicus* (f) | 86.92 | 6.63 | 0.16 | 3.26 | – | *Vergara & Rodriguez (2016)* |
| *P. californicus* (d) | 5.50 | 68.40 | 5.30 | 12.18 | – | *Bechtel et al. (2013)* |
| - (h) | – | 50.27 | 22.16 | – | – | *Babji et al. (2020)* |
| - (fr) | 76.5 | 9.3 | 1.3 | 12.4 | – | *Kim & Kim (2014)* |

**Note:**
–, Not determine; ad, air-dried; d, dried; f, fresh; fr, fermented; h, hydrolysate.

The reproductive success of sea cucumbers relies on their social behaviour, population diversity and density, chemical communication, and egg-laying synchronization (*Scannella et al., 2022*). A reduction in population density can have an impact on fertilization and lead to a decrease in population size, thus affecting their reproductive success (*Hasan, 2019*; *Gonzales-Duran et al., 2021*). Holothurians are vulnerable to changes in pH, temperature, and salinity (*Liu, 2015*; *Gonzales-Duran et al., 2021*), which can impact their population frequency, growth, and survival rates (*Kashenko, 2000*; *Dong et al., 2011*; *Liu, 2015*; *Zhou, Zhang & Li, 2018*). Additionally, larval development and

**Table 2 Fatty acid content of sea cucumber and sea cucumber viscera (%).**

| Species | SFA | MUFA | PUFA | EPA | DHA | References |
|---|---|---|---|---|---|---|
| **Whole sea cucumber** | | | | | | |
| A. mauritiana | 39.62 | 28.27 | 32.12 | – | – | Haider et al. (2015) |
| A. mollis | 23.62 | 22.61 | 53.75 | 10.63 | 4.64 | Liu et al. (2017) |
| A. japonicus | 56.00 | 23.15 | 18.5 | 1.44 | 7.8 | Anisuzzaman et al. (2019) |
| B. marmorata | 47.16 | 19.37 | 33.48 | 3.71 | 4.54 | Nishanthan et al. (2018) |
| Bohadschia sp. | 44.76 | 22.71 | 32.53 | 9.3 | 3.47 | Nishanthan et al. (2018) |
| H. arenicola | 15.91 | 33.17 | 50.92 | – | – | Haider et al. (2015) |
| H. edulis | 83.95 | – | 16.05 | – | – | Al Azad, Shaleh & Siddiquee (2017) |
| H. forskali | 22.95 | 6.44 | 43.64 | 10.49 | 1.01 | Santos et al. (2015) |
| H. mammata | 19.21 | 15.01 | 53.38 | 4.99 | 10.30 | Aydin et al. (2011) |
| H. scabra | 41.61 | 22.14 | 36.25 | 18.55 | 1.27 | Nishanthan et al. (2018) |
| H. spinifera | 40.12 | 30.72 | 29.16 | 10.95 | 0.61 | Nishanthan et al. (2018) |
| H. leucospilota | 34.55 | 30.14 | 35.29 | – | – | Yahyavi et al. (2012) |
| H. tubulosa | 15.48 | 13.29 | 57.76 | 6.18 | 12.37 | Aydin et al. (2011) |
| I. badionotus | 52.62 | 24.35 | 22.98 | 3.02 | 4.32 | Zacarias-Soto & Olvera-Novoa (2015) |
| S. chloronotus | 52.21 | 20.56 | 27.23 | 13.00 | 1.43 | Nishanthan et al. (2018) |
| T. anax | 45.02 | 29.47 | 25.51 | 5.51 | 0.99 | Nishanthan et al. (2018) |
| **Sea cucumber viscera** | | | | | | |
| A. chilensis | 50.39 | 23.97 | 6.42 | 2.71 | – | Careaga, Muniain & Maier (2013) |
| C. frondosa | – | – | 29.72 | 28.23 | – | Abuzaytoun et al. (2022) |
| C. frondosa | | | | | | Liu et al. (2021) |
| Air-dried | 26.46 | 28.73 | 40.14 | 28.71 | 0.87 | |
| Freeze-dried | 26.97 | 28.60 | 39.91 | 27.97 | 0.85 | |
| Fresh | 25.93 | 27.01 | 33.01 | 27.76 | 0.88 | |
| C. frondosa | 26.40 | 28.20 | 45.40 | 17.10 | 0.30 | Mamelona, Louis & Pelletier (2010) |
| C. frondosa | 3.19 | 11.3 | – | – | – | Ramalho et al. (2020) |
| P. californicus | 18.13 | 32.05 | 43.64 | 22.63 | 8.93 | Bechtel et al. (2013) |
| – | 39.76 | 26.86 | 33.39 | – | – | Babji et al. (2020) |

**Note:**
–, Not determined.

survival are highly susceptible to climate change (Asha & Muthiah, 2005; Zamora & Jeffs, 2013; Liu, 2015; Gonzales-Duran et al., 2021). Fluctuations in these environmental variables trigger a variety of biochemical and physiological adaptations that delay larval metamorphosis, especially during the gastrulation stage, which marks the development of digestive tract feeding activity. This results in slower larval development, increases the risk of predation, and adversely affects larval survival (Asha & Muthiah, 2005; Gonzales-Duran et al., 2021).

## Nutritional value of sea cucumber

The consumption of sea cucumbers as food products, tonics and aphrodisiacs is only popular in China and other Asian countries (Han, Keesing & Liu, 2016). However, sea

**Table 3 Amino acid profiles of sea cucumber and sea cucumber viscera (%).**

| Species | Leu* | His* | Lys* | Arg | Val* | Ile* | Thr* | Phe* | Met* | Tyr | Asp | Ala | Pro | Gly | Gln | Ser | Cys | References |
|---|---|---|---|---|---|---|---|---|---|---|---|---|---|---|---|---|---|---|
| **Whole sea cucumber** | | | | | | | | | | | | | | | | | | |
| A. mauritiana | 1.58 | 0.65 | 3.52 | 0.99 | 2.13 | 0.43 | 2.19 | 0.99 | 0.42 | 0.33 | 4.48 | 6.45 | 0.24 | 18.80 | 5.25 | 2.11 | – | Omran (2013) |
| H. arenicola | 5.19 | 1.41 | 2.06 | 6.12 | 2.94 | 3.37 | 4.59 | 2.80 | 0.43 | 2.45 | 15.71 | 11.72 | 7.56 | 17.33 | 11.77 | 4.54 | – | Haider et al. (2015) |
| H. leucospilota | 1.86 | 0.36 | 0.73 | 1.71 | 1.51 | 0.49 | 2.73 | 0.75 | 0.19 | 0.49 | 4.65 | 5.80 | 0.14 | 19.17 | 5.64 | 2.33 | – | Omran (2013) |
| H. mammata | 3.70 | 2.00 | 2.20 | 6.80 | 3.70 | 2.00 | 3.30 | 3.30 | 1.00 | 7.10 | 5.30 | 12.20 | 8.20 | 11.80 | 12.10 | 3.20 | 8.20 | Gonzales-Wanguemert et al. (2018) |
| H. polii | 5.40 | – | 1.10 | 13.40 | 5.40 | 2.90 | 3.40 | 8.10 | 0.70 | 3.80 | 4.50 | 15.10 | 10.10 | 10.60 | 8.40 | 3.00 | 2.40 | Gonzales-Wanguemert et al. (2018) |
| H. scabra | 2.33 | 0.46 | 1.91 | 4.37 | 1.85 | 1.36 | 2.52 | 1.02 | 0.66 | 1.13 | 5.46 | 4.52 | 6.29 | 8.22 | 8.32 | 2.20 | 6.47 | Doungpai et al. (2022) |
| H. tubulosa | 5.80 | – | 1.60 | 12.90 | 5.50 | 3.30 | 3.70 | 7.90 | 0.10 | 3.30 | 4.80 | 14.50 | 9.50 | 10.60 | 8.60 | 3.50 | 2.40 | Gonzales-Wanguemert et al. (2018) |
| S. horrens | 1.27 | 0.30 | 0.47 | 3.76 | 1.07 | 0.87 | 1.51 | 0.96 | 0.33 | 0.75 | 2.66 | 3.09 | 3.04 | 8.29 | 4.89 | – | – | Stiani et al. (2021) |
| **Sea cucumber viscera** | | | | | | | | | | | | | | | | | | |
| C. frondosa | 0.78 | 1.04 | 0.59 | – | 0.17 | 15.56 | 10.00 | 15.31 | 8.12 | 1.07 | 0.50 | 4.92 | 10.80 | 3.17 | 25.93 | 1.70 | 0.35 | Liu et al. (2021) |
| C. frondosa | 7.20 | 2.30 | 6.60 | 9.10 | 5.40 | 4.70 | 5.00 | 3.50 | 2.30 | 5.00 | 10.00 | 6.60 | 4.00 | 8.00 | 14.30 | 4.30 | 1.60 | Mamelona, Louis & Pelletier (2010) |
| – | 4.24 | 1.24 | 1.11 | 5.24 | 2.36 | 2.15 | 4.19 | 2.23 | 1.08 | 2.06 | 5.64 | 2.18 | 2.75 | 4.22 | 7.36 | 3.59 | 0.01 | Babji et al. (2020) |

**Note:**

Leu, leucine; His, histidine; Lys, lysine; Arg, arginine; Val, valine; Ile, isoleucine; Thr, threonine; Phe, phenylalanine; Met, methionine; Tyr, tyrosine; Asp, aspartic acid; Pro, proline; Gly, glycine; Gln, glutamine; Ser, serine; Cys, cysteine; *, essential amino acid; –, not determined.

**Table 4 Mineral compositions of sea cucumber and sea cucumber viscera (g/100 g).**

| Species | Ca | Na | Mg | P | K | Fe | References |
|---|---|---|---|---|---|---|---|
| **Sea cucumber** | | | | | | | |
| H. arenicola | 0.083 | – | 0.115 | – | – | 0.060 | Barzkar, Fariman & Taheri (2017) |
| H. sanctori | 0.657 | 0.552 | 0.156 | 0.011 | – | – | Gocer, Olgunoglu & Olgunoglu (2018) |
| H. scabra | 1.821 | 0.666 | 0.305 | 0.088 | 0.061 | 0.022 | Rasyid et al. (2020) |
| H. tubulosa | 2.807 | 3.902 | 0.431 | 0.048 | 0.443 | <0.001 | Kunili & Colakoglu (2019) |
| P. californicus | 0.002 | 0.008 | 0.001 | <0.001 | <0.001 | 0.018 | Bechtel, Oliveira & Smiley (2012). |
| P. parvimensis | 0.095 | 0.016 | 0.011 | 0.014 | 0.047 | 0.021 | Chang-Lee, Price & Lampila (1989) |
| S. horrens | 0.106 | – | 0.093 | – | – | 0.521 | Barzkar, Fariman & Taheri (2017) |
| S. japonicus | 0.003 | – | <0.001 | – | – | – | Tanikawa (1955) |
| S. vastus | 2.449 | 8.054 | – | 5.085 | 0.160 | 0.521 | Rasyid (2017) |
| **Sea cucumber viscera** | | | | | | | |
| C. frondosa | 0.900 | 1.240 | – | – | 1.870 | 0.019 | Mamelona, Louis & Pelletier (2010) |
| C. frondosa | 0.014 | 0.190 | – | 0.110 | 0.200 | 0.009 | Ramalho et al. (2020) |
| P. californicus | – | – | – | – | – | 0.003 | Bechtel et al. (2013) |
| – | 128.3 | 338.3 | – | – | 0.05 | 39.70 | Babji et al. (2020) |
| – | 0.075 | – | – | 0.112 | – | – | Kim & Kim (2014) |

**Note:**

–, Not determined.

cucumbers are very nutritious, and the bioactive components found in them are valuable for both the food and the biomedical industries (*Pangestuti & Arifin, 2018*). The body wall is the main part of the sea cucumber that is consumed and consists mainly of epithelial and dermal connective tissues, collagenous fibres, proteoglycans, glycoproteins, and amorphous interstitial materials (*Yang, Hamel & Mercier, 2015*).

The proximate compositions (moisture, protein, lipid, ash, and carbohydrates) of various sea cucumber species are shown in Table 1. Most sea cucumber species exhibit high amounts of saturated fatty acids (SFAs) and monounsaturated fatty acids (MUFAs) (Table 2) and are abundant in essential amino acids (Table 3). SFAs were found to be the dominat fatty acid elements in *H. scabra* (71.76%), *H. leucospilota* (69.57%), and *H. atra* (57.04%) (*Ridzwan et al., 2014*).

Additionally, sea cucumbers have outstanding vitamin and mineral profiles, including vitamins A (455 µg/100 g), B1 (thiamine) (0.04 mg/kg), B2 (riboflavin) (0.06 mg/kg), B3 (niacin) (0.4 mg/kg), C (3.19 mg/100 g), and E (2.82 mg/100 g), as well as essential minerals, particularly calcium, magnesium, and iron (*Bordbar, Anwar & Saari, 2011*; *Sroyraya et al., 2017*; *Achmad et al., 2020*; *Ardiansyah et al., 2020*) (Table 4). Notably, *Stichopus vastus* had the highest mineral content compared to other species. In addition to being valuable marine commodities, sea cucumbers are a significant source of medicine (*Zhao et al., 2018*). They are used in traditional Chinese medicine and are thought to have therapeutic capabilities, to treat conditions such as arthritis, high blood pressure, asthma, cancer, frequent urination, and impotence (*Guo et al., 2015*; *Pangestuti & Arifin, 2018*; *Liang et al., 2022*). They have extensive uses in the biomedical industry, where they are believed to possess therapeutic capabilities (*Zohdi et al., 2011*) and numerous active ingredients, including polysaccharides, peptides, proteins, lipids, (*Janakiram, Mohammed & Rao, 2015*), collagen, gelatine, saponins and acid mucopolysaccharides (*Kariya et al., 2004*; *Lu et al., 2010*; *Zhou, Wang & Jiang, 2012*; *Yang, Hamel & Mercier, 2015*).

## Socio-economic status of sea cucumber

Monetized marine resources, such as sea cucumbers, greatly support the livelihoods of coastal communities in the Indo-Pacific region (*Hair et al., 2019*). Sea cucumbers are easily processed (gutted, boiled, and dried) using basic, affordable tools (*Ram et al., 2017*; *Hair et al., 2019*). They are relatively sedentary, making them easy to collect (*Wolfe & Byrne, 2022*). They are commonly harvested by fisher in traditional ways, such as collecting from coral reefs at low tide or by diving in shallow waters (*Friedman et al., 2008*; *Plaganyi et al., 2020*; *Prasada, 2020*).

Coastal communities process sea cucumbers and sell the final products at a higher price. The processed and dried body wall of sea cucumbers, known as 'beche-de-mer', is a valuable marine export commodity and an important source of income for coastal communities. It is massively exported throughout Asia and is highly regarded as a seafood delicacy. Prices for sea cucumbers can fetch up to US$983.47 per kilogram in China or US$110.78 per kilogram in Japan (*Kinch et al., 2007*; *Purcell, Williamson & Ngaluafe, 2018*; *Hair et al., 2019*; *Wolfe & Byrne, 2022*). The market price on the market is not only based on the type and species but also on size. There are various grades of processed sea

cucumbers in demand, but in general, sellers often grade sea cucumbers into at least three grades: grade 1 (largest size), grade 2 (medium size) and grade 3 (smallest size) (*Ochiewo et al., 2010*).

Greater quantities of 'beche-de-mer' are exported to Asia, where it is a valuable product. *Louw & Burgener (2020)* reported Madagascar is the top exporter of dried sea cucumbers, sending 40% of its total exports to Asian markets between 2012 and 2019. Hong Kong serves as a major importer and commercial hub for dried seafood, including sea cucumber (*Ben-Hasan et al., 2021*). Over the past 8 years, Hong Kong has been the greatest importer in Asia, bringing in almost 56 million kg from around the world. However, nearly 70% of its total imports are then re-exported to mainland China (*Jun, 2009*; *Louw & Burgener, 2020*; *Ben-Hasan et al., 2021*). Due to the high demand in the Asian market and the high shipping costs that restrict reef fish exports, sea cucumber fisheries have become the second-most profitable export fishery in the South Pacific, after tuna (*Carleton et al., 2013*).

# NUTRITIONAL VALUE OF SEA CUCUMBER VISCERA

## Novel sulphated polysaccharide

The viscera of sea cucumbers are a good source of nutrients. The purified chondroitin sulphate polysaccharides from the digestive tract of *Apostichopus japonicus* are postulated to have anti-tumour proliferation properties (*Wei, Jian & Mao, 2011*; *Xin, Jing & Jian, 2016*), whereas the purified sulphated polysaccharides from *Pattalus mollis* have an anticoagulant effect (*Zheng et al., 2019*). Sea cucumber viscera contain approximately 4.9% crude polysaccharides, from which sulphated polysaccharides can be extracted (*Yang et al., 2020*). Sulphated polysaccharides have a variety of beneficial biological properties, including anticoagulant, antiviral, antioxidant, anticancer, and immuno-inflammatory properties, making them suitable for nutraceutical, cosmeceutical, and pharmaceutical applications (*Wijesekara, Pangestuti & Kim, 2011*; *Jiao et al., 2011*; *Zhu et al., 2018*).

## Novel saponins

Sea cucumber viscera are abundant in saponins—secondary metabolites that influence metabolism and enhance the immune system by reducing cholesterol and exhibiting anti-cancer (*Shi et al., 2004*) and anti-bacterial properties (*Sumarto & Karnila, 2022*). Recent studies revealed that viscera contain higher saponin content than their body walls (*Zhang et al., 2018*). Novel saponins with potent fungicidal, antioxidant, anti-viral, and anti-cancer effects have been identified and purified from the viscera of *Holothuria scabra*, *Holothuria lessoni* and *Apostichopus japonicus*, making them promising candidates for cosmeceutical, medical, and pharmaceutical applications (*Bahrami, Zhang & Franco, 2014*; *Zhang et al., 2018*; *Sumarto & Karnila, 2022*).

## Novel source of nutrients

Numerous nutrients, including oligosaccharides, phenols, flavonoids, and trace metals, have been found in the viscera of sea cucumbers (*Zhang & Chang, 2014*). the reported proximate composition of sea cucumber viscera may differ among studies (Table 1) due to species, sample preparation techniques, feed, and environmental factors such as

**Table 5 Vitamin contents in sea cucumber viscera (mg/kg).**

| Parameter | Species | |
|---|---|---|
| | **Undetermined species** | *Cucumaria frondosa* |
| Vitamin B1 (Thiamine) | 80 | 0.439 |
| Vitamin B2 (Riboflavin) | 458 | 1.081 |
| Vitamin B3 (Nicotinic acid) | 947 | 6.704 |
| Vitamin B6 (Pyridoxine) | 60 | – |
| Vitamin B5 (Pantothenic acid) | 170 | 3.157 |
| Vitamin B7 (Biotin) | 0.2 | – |
| Vitamin B9 (Folic acid) | 0.107 | 0.193 |
| Vitamin B12 (Cobalamin) | 1.222 | – |
| Vitamin C | 55 | – |
| Vitamin E as a-tocopherol | 1.4 | 1.947 |
| Vitamin A as b-carotene | 23 | – |
| Vitamin A as retinol | – | 0.015 |
| Vitamin D | 0.037 | – |
| Vitamin K | 1.2 | – |
| References | *Babji et al. (2020)* | *Mamelona, Louis & Pelletier (2010)* |

**Note:**
–, Not determine.

habitat and climate. Sea cucumber viscera contain low lipid but high protein contents. *He et al. (2017)* demonstrated that protein hydrolysate from sea cucumber viscera had high antifatigue and antioxidant activity. Additionally, due to their high profile of essential amino acids, sea cucumber protein hydrolysate may be a suitable dietary protein source (*Senadheera, Dave & Shahidi, 2021*).

*Babji et al. (2020)* reported that sea cucumber viscera contain high levels of vitamins. The consumption of a relatively small amount of sea cucumber viscera may satisfy the nutritional needs for several vitamins such as vitamins A, B, C, and E in both animal and human diets, which support the immune system, strengthen bones, heal wounds, turn food into energy, and repair cellular damage. The vitamin content in sea cucumber viscera is shown in Table 5. According to Table 5, vitamin B3 (nicotinic acid) is the most abundant vitamin in the viscera of both *C. frondosa* (*Mamelona, Louis & Pelletier, 2010*) and undetermined species (*Babji et al., 2020*).

Furthermore, sea cucumber viscera contain a significant number of essential minerals (*e.g.*, Cu, Fe, Zn, K, Na, Mn, As, Mg, Se, Ni, and Ca) that play important roles in maintaining general biological systems, along with relatively small amounts of nonessential trace elements (Cd, Co, and Pb) (*Mamelona, Louis & Pelletier, 2010*). The mineral composition of sea cucumber viscera is shown in Table 4. *Babji et al. (2020)* reported that sodium is the highest element content in sea cucumber viscera, approximately 338.3 g/100 g, while potassium was found to be the most abundant element in *Cucumaria frondosa*, at approximately 1.870 g/100 g, as reported by *Mamelona, Louis & Pelletier (2010)*, and 0.200 g/100 g, as reported by *Ramalho et al. (2020)*. Potassium is a

micromineral that regulates the activity of blood cells and muscles, especially the heart muscle, maintains fluid balance in the body, regulates blood pressure, and acts as an enzyme activator (*Susanti, Sukmawardani & Musfiroh, 2016*).

Sea cucumber viscera are rich in fatty acids, especially eicosapentaenoic acid (EPA) and docosahexaenoic acid (DHA), which are essential micronutrients with beneficial health impacts (*Shepon et al., 2022*). In addition, *Mamelona, Louis & Pelletier (2010)* reported that sea cucumber viscera are abundant in essential amino acids, which play a crucial role in metabolic pathway regulation and are crucial components in growth, development, and reproduction. The fatty acid and amino acid profiles of sea cucumber viscera reported by several studies are shown in Tables 2 and 3, respectively. *Liu et al. (2021)* highlighted that EPA was the predominant fatty acid, constituting 27.76% of the total fatty acids found in the viscera of *Cucumaria frondosa*. This percentage is significantly higher than that of crude oil from other marine byproducts, which typically contain only 4.63–9.54% EPA. EPA, as an essential omega-3 fatty acid, has been linked to numerous positive health effects, including decreased cardiovascular risk, stimulation of foetal development, and improved cognitive function (*Swanson, Block & Mousa, 2012*). Therefore, sea cucumber viscera can be valorised to produce omega-3 PUFAs (especially EPA) and EPA-enriched nutritional products. Additionally, approximately 25.93% (*Liu et al., 2021*) and 14.30% (*Mamelona, Louis & Pelletier, 2010*) of the total amino acids in the viscera of *Cucumaria frondosa* are composed of glutamine, which is essential for cellular metabolic activities and animal immunity (*Binod et al., 2017*; *Shah, Wang & Ma, 2020*). The nutritional content of sea cucumber viscera and the richness of their metabolites indicates their enormous potential for being transformed into high-value products.

Due to their high nutritional value sea cucumber viscera have traditionally been consumed both raw and processed in some countries. In Samoa, *Stichopus horrens* is the most sought-after species, as the viscera are consumed raw and commercially sold in local markets, while the body parts are returned to the sea alive (*Eriksson et al., 2007*; *Charan-Dixon et al., 2019*). In addition, salt-fermented sea cucumber viscera, known as konowata, is one of the three major Japanese delicacies that are commercially consumed in Japan (*Nishanthan et al., 2021*). Their edibility and nutritional value indicate their potential to be consumed as therapeutic foods.

## FUTURE DIRECTIONS

### Valorisation of sea cucumber viscera into value-added products

Sea cucumber viscera contain sulphated polysaccharides (*e.g.*, sulphate, fucose, galactosamine) that can be used in pharmacological activities, such as improving gut health, antiviral mechanisms and wound healing (*Cao, Surayot & You, 2017*; *Li et al., 2021*). Conventional extraction methods, such as chemical methods can be applied to extract sulphated polysaccharides from sea cucumber viscera. However, the alkaline conditions during the chemical extraction process can influence the conformation of sulphated polysaccharides, leading to potential degradation and desulphation of the polysaccharides. Consequently, this may affect the physicochemical and biological properties of the polysaccharides (*Li et al., 2021*).
Due to the limitations of conventional extraction methods, microwave-assisted extraction has emerged as a promising technology to minimize degradation and desulphation during the extraction of sulphated polysaccharides from sea cucumber viscera. The use of microwave heating can lead to a reduction in the use of solvents, thereby lowering operating costs. Moreover, microwave radiation provides rapid heating, which, in turn, reduces energy consumption during extraction (*Wan Mahari et al., 2022*). The microwave radiation generated during heat activation can contribute to cell wall lysis and break the protein bonds to release the cellular contents and polysaccharides into the extraction medium. However, further studies are needed to optimize the operating conditions (*e.g.*, microwave power, duration) during microwave-assisted extraction to enhance the yield and properties of sulphated polysaccharides.

Enzymatic hydrolysis is another emerging technology that can be applied to improve the extraction of sulphated polysaccharides (*Wang et al., 2022*). The types of proteases (*e.g.*, papain, pancreatin) and duration of hydrolysis are important factors that influence the release of polysaccharides from sea cucumber viscera and their biological properties. Previous study reported that protein hydrolysates produced by pancreatin exhibited greater antioxidant activity compared to that produced by papain (*Karamac, Kosińska-Cagnazzo & Kulczyk, 2016*). Therefore, further studies should be scrutinized to develop an extraction method that can enhance the production of sulfated polysaccharides with desirable physicochemical and biological properties.

## Potential cosmetic ingredients from sea cucumber viscera

The abundance of essential active compounds found in sea cucumber viscera extracts highlights their high potential for use in the cosmeceutical field. Studies have reported that sea cucumber (*S. japonicus*) viscera extracts promote the expression of tyrosinase, tyrosinase-related protein (TRP-1 and TRP-2), and microphthalmia-associated transcription factor (MITF) protein levels, as well as extracellular-regulated kinase (ERK) activation. These elements are known to be effective in skin whitening and anti-ageing treatments by reducing melanin production and increasing collagen synthesis through ERK signalling (*Kwon et al., 2018*).

## Potential as animal nutrition supplements and feed additives

In aquaculture, good nutrition is essential for the optimal growth and production of high-quality products, and the feed component typically represents approximately 60–80% of the production cost (*Ragasa, Osei-Mensah & Amewu, 2022*). Feed additives are products used to enhance nutrients and reduce production costs. Functional feed additives in animal food stimulate growth, promote good health, boost the immune system, and provide physiological advantages over conventional feeds (*Alemayehu, Geremew & Getahun, 2018*).

Sea cucumber viscera are a high-value source of nutrients and bioactive compounds that can be used as functional feed additives in the aquaculture industry. *Babji et al. (2020)* successfully converted sea cucumber viscera into prospective health supplement products for the agro-based food and health industries using an enzymatic hydrolysis method. This

review demonstrates the potential of turning waste materials into high-value products, which not only increases the market value of sea cucumber viscera but also helps reduce waste that could otherwise pollute the environment.

## Potential as sex reversal agent for aquatic animal

The males of many species of ornamental fish have highly pigmented bodies and usually more developed fins, making them preferred over female fish by hobbyists (*Piferrer & Lim, 1997*). Due to the higher demand for these male fish, the use of sex reversal technology for commercial production of an all-male population of these aquatic animals could significantly increase the economic benefits of this type of aquaculture operation. The hormones used for sex reversal to produce all-male fish are testosterone, 17-α-methyltestosterone, and androstenedione. In aquaculture, sex reversal can be triggered by using steroid hormones either through immersion, injection, or oral administration by feeding. However, synthetic steroids have a negative impact on the environment and the fish itself (*Emilda, 2015*). Therefore, it is essential to explore natural steroid sources that are safe for both humans and the environment.

Testosterone, the steroid hormone used in sex reversal, is naturally produced in sea cucumbers. The potential utilization of sea cucumber extract as a natural source of testosterone is promising. However, the optimal extraction method has yet to be determined. Recently, studies have been conducted on sex reversal in aquatic animals using steroid extracts from sea cucumber viscera. *Emilda (2015)* and *Saputra et al. (2018)* reported that using the immersion technique to administer steroid extracts from sea cucumber viscera can enhance the percentage of male guppies by 65.13% and 88.9%, respectively. *Riani, Sudrajat & Triajie (2010)* reported that injecting testosterone hormone from sea cucumber viscera into giant freshwater prawns can produce 63.33% male prawns. The hormone is also effective in influencing zygotes and larvae to develop into male prawns without negatively impacting motility, fecundity, and hatching rates. This research provides crucial baseline information, indicating that sea cucumber viscera can serve as a natural steroid source for sex reversal in aquatic organisms.

## Promising neuroprotective agent in debilitating central nervous system disorders

Triterpene glycoside or saponins are natural bioactive compounds found in sea cucumbers (*Bahrami, Zhang & Franco, 2018*). Numerous studies have demonstrated that the medicinal and health benefits of sea cucumbers are attributed to the presence of saponins, which are the most significant and primary secondary metabolites found in sea cucumbers (*Zhao et al., 2018*). Saponins have neuroprotective properties against a number of central nervous system disorders (*Mitu et al., 2017*).

*Kleawyothatis et al. (2022)* reported that the whole body (including viscera) and body wall extracts of sea cucumber (*H. scabra*) contain triterpene glycosides or saponins. These extracts have been shown to provide a neuroprotective effect against central nervous system disorders, specifically Alzheimer, in *Caenorhabditis elegans* models. The observed decrease in amyloid-β deposition and aggregation, along with the reduction in reactive

oxygen species, resulted in lifespan extension of *C. elegans* Alzheimer's disease models. These findings strongly suggest that sea cucumber extracts possess natural preventive and therapeutic properties for Alzheimer's disease. Meanwhile, *Bahrami, Zhang & Franco (2018)* mentioned that sea cucumber viscera had substantially greater relative levels of saponins than the body and also contained some saponin congners. Although saponins are known to have significant neuroprotective properties (*Mitu et al., 2017*), further studies are needed to validate the role of saponins found in sea cucumber viscera in preventing and treating debilitating central nervous system disorders.

## CONCLUSIONS

Sea cucumber viscera are excellent sources of nutrients and bioactive compounds, including saponins, sulphated polysaccharides, amino acids, fatty acids (especially EPA and DHA), vitamins, and minerals. These compounds can be developed into potential animal nutritional supplements and feed additives, utilized as sex reversal agents for aquatic animals, and incorporated into various industries, such as food, cosmeceutical, and biopharmaceutical. The potential utilization of sea cucumber viscera as a natural source of nutrients is promising, but the optimal extraction method has not yet been found, hindering their use on an industrial scale. Considering the abundance of various nutrients and bioactive compounds in sea cucumber viscera, further research should be conducted to optimize the extraction process and enable the industrial production of high-value sea cucumber viscera and support the development of sea cucumber aquaculture technology.

### Funding

This work was supported by the University Malaysia Terengganu and Hasanuddin University, Indonesia under the International Partnership Research Grant (UMT/IPRG/2021/55300). The funders had no role in study design, data collection and analysis, decision to publish, or preparation of the manuscript.

### Grant Disclosures

The following grant information was disclosed by the authors:
University Malaysia Terengganu and Hasanuddin University.
Indonesia under the International Partnership Research Grant: UMT/IPRG/2021/55300.

### Competing Interests

Khor Waiho is an Academic Editor for PeerJ.

### Author Contributions

- Muhammad Fatratullah Muhsin conceived and designed the experiments, prepared figures and/or tables, and approved the final draft.
- Yushinta Fujaya conceived and designed the experiments, authored or reviewed drafts of the article, and approved the final draft.

- Andi Aliah Hidayani performed the experiments, prepared figures and/or tables, and approved the final draft.
- Hanafiah Fazhan performed the experiments, prepared figures and/or tables, and approved the final draft.
- Wan Adibah Wan Mahari performed the experiments, prepared figures and/or tables, and approved the final draft.
- Su Shiung Lam performed the experiments, authored or reviewed drafts of the article, and approved the final draft.
- Alexander Chong Shu-Chien analyzed the data, authored or reviewed drafts of the article, and approved the final draft.
- Youji Wang analyzed the data, authored or reviewed drafts of the article, and approved the final draft.
- Nor Afiqah-Aleng analyzed the data, prepared figures and/or tables, and approved the final draft.
- Nita Rukminasari analyzed the data, authored or reviewed drafts of the article, and approved the final draft.
- Khor Waiho conceived and designed the experiments, prepared figures and/or tables, and approved the final draft.

## Data Availability

This is a literature review and hence did not utilize raw data.

## Supplemental Information

Supplemental information for this article can be found online at http://dx.doi.org/10.7717/peerj.16252#supplemental-information.

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

## FURTHER READING

**Abedin MZ, Karim AA, Ahmed F, Latiff AA, Gan Y, Ghazali FC, Sarker MZI. 2012.** Isolation and characterization of pepsin-solubilized collagen from the integument of sea cucumber (*Stichopus vastus*). *Journal of the Science of Food and Agriculture* **93(5)**:1083–1088 DOI 10.1002/jsfa.5854.

**Afkhami M, Ehsanpour M, Khazaali A, Dabbagh A, Maziar Y. 2012.** New observation of a sea cucumber, Holothuria (Thymiosycia) impatiens, from Larak Island (Persian Gulf, Iran). *Marine Biodiversity Records* **5(3)**:E62 DOI 10.1017/S1755267212000425.

**Ahmed Q, Thandar AS, Ali QM. 2020.** *Holothuria (Lessonothuria) insignis* Ludwig, 1875 (formally resurrected from synonymy of *H. pardalis* Selenka, 1867) and *Holothuria (Lessonothuria) lineata* Ludwig, 1875-new additions to the sea cucumber fauna of Pakistan, with a key to the subgenus *Lessonothuria* Deichmann (Echinodermata: Holothuroidea). *Zootaxa* **4767(2)**:307–318 DOI 10.11646/zootaxa.4767.2.6.

**Amiri M, Hosseini H, Islami HR. 2013.** Identification and distribution of sea cucumbers in the Chabahar Bay, Iran. In: *VI International Conferences "Water & Fish"*. 487–490.

**Bruckner AW, Johnson KA. 2003.** Conservation strategies for sea cucumber: can a CITES Appendix II listing promote sustainable international trade? *SPC Beche-de-mer Information Bulletin* **18**:24–33.

**Canada MCB, Resueno MA, Angara EV. 2020.** Species distribution, diversity, and abundance of sea cucumber in tropical intertidal zones of aurora, Philippines. *Journal of Ecology* **10(12)**:1–10 DOI 10.4236/oje.2020.1012047.

**Cannon LRG, Silver H. 1986.** Sea cucumber of Northern Australia. Queensland Museum, South Brisbane, Australia60. *Available at https://catalogue.nla.gov.au/catalog/2031103*.

**Chen X, Sun Y, Zhao H, Hu J, Chen B, Li H, Huang W. 2021.** Complete mitochondrial genome of a tropical sea cucumber, *Stichopus chloronotus. Mitochondrial DNA B Resource* **6(10)**:2788–2790 DOI 10.1080/23802359.2021.1967218.

**Clark AM, Rowe FW. 1971.** *Monograph of swallow water Indo-West Pacific Echinoderms*. London, UK: Trustees of the British Museum (Natural History), 238.

**Conand C, Purcell S, Gamboa R. 2013.** *Thelenota anax*. The IUCN Red List of Threatened Species 2013: e.T180324A1615023. *Available at https://www.iucnredlist.org/species/pdf/1615023*.

**Dissanayake DCT, Athukorala S. 2011.** Abundance, distribution and some biological aspects of *Holothuria edulis* off the northwest coast of Sri Lanka. *SPC Beche-de-mer Information Bulletin* **31**:39–44.

**Dolorosa RG, Salazar CL, Delfin MTV, Paduga JR, Balisco RAT. 2017.** Sea cucumber fisheries in Rasa Island Wildlife Sanctuary, Narra, Palawan, Philippines. *SPC Beche-de-mer Information Bulletin* **37**:9–20.

**Dong P, Xue C, Du Q. 2008.** Separation of two main triterpene glycosides from sea cucumber *Pearsonothuria graeffei* by high-speed countercurrent chromatography. *Acta Chromatographica* **20(2)**:269–276 DOI 10.1556/achrom.20.2008.2.11.

**Eriksson H, Byrner M, Torre-Castro MDL. 2012.** Sea cucumber (Aspidochirotida) community, distribution and habitat utilization on the reefs of Mayotte, Western Indian Ocean. *Marine Ecology Progress Series* **452**:159–170 DOI 10.3354/meps09665.

**Ghanbari R, Ebrahimpour A, Abdul-Hamid A, Ismail A, Saari N. 2012.** *Actinopyga lecanora* hydrolysate as natural antibacterial agents. *International Journal of Molecular Science* **13(12)**:16796–16811 DOI 10.3390/ijms131216796.

**Hamel J, Conand C, Pawson DL, Mercier A. 2001.** The sea cucumber *Holothuria scabra* (Holothuroidea: Echinodermata): its biology and exploitation as beche-de-mer. *Advanced in Marine Biology* **41**:129–223 DOI 10.1016/S0065-2881(01)41003-0.

**Hamel J, Mercier A. 1996.** Early development, settlement, growth, and spatial distribution of the sea cucumber *Cucumaria frondosa* (Echinodermata: Holothuroidea). *Canadian Journal of Fisheries and Aquatic Science* **53(2)**:253–271 DOI 10.1139/f95-186.

**Herath HMTB, Hewage HHP, Nirooparaj B, Senanayake SAMAIK. 2020.** First record of Lampert's sea cucumber (*Synaptula lamperti*) in Northern Coast, Sri Lanka. In: *International Conferences of Multidisciplinary Research*. 375–378.

**Hossain A, Dave D, Shahidi F. 2020.** Northern sea cucumber (*Cucumaria frondose*): a potential candidate for functional food, nutraceutical, and pharmaceutical sector. *Marine Drugs* **18(5)**:274 DOI 10.3390/md18050274.

**Idressbabu KK, Sureshkumar S. 2017.** Distribution pattern and community structure of sea cucumbers (Class: Holothuroidea) in different biogeographic regions of the selected Island of Lakshadweep Archipelago, India. *Indian Journal of Geo Marine Sciences* **46(3)**:569–575.

**Khazaali A, Kunzmann A, Bastami KD, Baniamam M. 2016.** Baseline of polycyclic aromatic hydrocarbons in the surface sediment and sea cucumbers (*Holothuria leucospilota* and *Stichopus hermanni*) in the northern parts of Persian Gulf. *Marine Pollution Bulletin* **110(1)**:539–545 DOI 10.1016/j.marpolbul.2016.05.039.

**Lane DJW. 1999.** Distribution and abundance of *Thelenota rubralineata* in the western Pacific: some conservation issues. *SPC Beche-de-mer Information Bulletin* **11**:19–21.

**Lane DJW, Vandenspiegel D. 2003.** *A guide to sea stars and other Echinoderms of Singapore.* Singapore. Singapore Science Centre, 187.

**Lee T, Shin S. 2019.** Complete mitochondrial genome of sea cucumber, *Holothuria (Stauropora) pervicax* (Holothuroidea, Holothuriida, Holothuriidae), from Jeju Island, Korea. *Mitochondrial DNA Part B* **4(1)**:1047–1048 DOI 10.1080/23802359.2019.1584058.

**Liu H, Shen M. 2021.** Characterization and phylogenetic analysis of the complete mitochondrial genome of prickly redfish *Thelenota ananas* (Jaegar, 1833). *Mitochondrial DNA Part B* **6(8)**:2275–2277 DOI 10.1080/23802359.2021.1920509.

**Massin C. 1996.** Result of the rumphius biohistorical expedition to Ambon (1990) part 4. The Holothurioidea (Echinodermata) collected at Ambon during the rumphius biohistorical expedition. *Zoologische Verhandelingen* **307**:1–53.

**Massin, Zulfigar Y, Hwai ATS, Boss SZR. 2002.** The genus *Stichopus* (Echinodermata: Holothuroidea) from the Johore marine park (Malaysia) with the description of two new species. *Biologie* **72**:73–99.

**Mercier A, Battaglene SC, Hamel J. 2000.** Periodic movement, recruitment and size-related distribution of the sea cucumber *Holothuria scabra* in Solomon Island. *Hydrobiologia* **440**:81–100 DOI 10.1023/A:1004121818691.

**Mezali K, Thandar AS, Khodja I. 2021.** On the taxonomic status of *Holothuria (Holothuria) tubulosa* (s.s) from the Algerian coast with the description of a new Mediterranean species,

*Holothuria* (*Holothuria*) *algeriesis* n. sp. (Echinodermata: Holothuroidea: Holothuriidae). *Zootaza* **4981(1)**:89–106 DOI 10.11646/zootaxa.4981.1.4.

**Minami K, Sawada H, Masuda R, Takaahashi K, Shirawaka H, Yamashita Y. 2018.** Stage-specific distribution of Japanese sea cucumber *Apaostichopus japonicus* in Maizuru Bay, Sea of Japan, in relation to environmental factors. *Fisheries Science* **84**:251–259 DOI 10.1007/s12562-017-1174-1.

**Mosher C. 1980.** Distribution of *Holothuria Arenicola* Semper in the Bahamas with observation on the habitat, behavior, and feeding activity (Echinodermata: Holothuroidea). *Bulletin of Marine Science* **30(1)**:1–12.

**O'Loughlin PM, Barmos S, VandenSpiegel D. 2011.** The paracaudinid sea cucumbers of Australia and New Zealand (Echinodermata: Holothuroidea: Molpadida: Caudinidae). *Memoirs of Museum Victoria* **68(1)**:37–65 DOI 10.24199/j.mmv.2011.68.03.

**Ongkers OTS, Pattinasarany M, Mamesah JAB, Uneputty PRA, Pattikawa JA. 2018.** Size distribution and growth pattern of *Holothuria atra* and *Holothuria scabra* in the coastal waters of Morella, Central Maluku Indonesia. *International Journal of Fisheries and Aquatic Studies* **6(5)**:301–305.

**Purcell SW, Piddocke TP, Dalton SJ, Wang Y. 2016b.** Movement and growth of the coral reef holothuroids *Bohadschia argus* and *Thelenota ananas*. *Marine Ecology Progress Series* **551**:201–214 DOI 10.3354/meps11720.

**Purcell SW, Samyn T, Conand C. 2012.** Commercially important sea cucumbers of the world. FAO Species Catalogue for Fishery Proposes No.6. *Available at* https://www.fao.org/3/i1918e/i1918e.pdf.

**Rakaj A, Fianchini A, Boncagni P, Lovatelli A, Scardi M, Cataudella S. 2017.** Spawning and rearing of *Holothuria tubulosa*: a new candidate for aquaculture in the Mediterranean region. *Aquaculture Research* **49(1)**:557–568 DOI 10.1111/are.13487.

**Reichenbach N. 1999.** Ecology and fishery biology of *Holothuria fuscolgiva* in the Maldives, Indian Ocean (Echinodermata: Holothuroidea). *Bulletin of Marine Science* **64(1)**:103–113.

**Sicuro B, Levine J. 2011.** Sea cucumber in the Mediterranean: a potential species for aquaculture in the Mediterranean. *Review in Fisheries Science* **19(3)**:299–304 DOI 10.1080/10641262.2011.598249.

**Siddique S, Ayub Z. 2015.** Population dynamics and reproduction of *Holothuria arenicola* (Holothuroidea: Echinodermata) in coastal waters of Pakistan, North Arabian Sea. *Journal of the Marine Biological Association of the United Kingdom* **95(6)**:1–10 DOI 10.1017/S0025315415000041.

**Skewes TD, Murphy NE, McLeod I, Dovers E, Burridge C, Rochester W. 2010.** Torres strait hand collectables, 2009 survey: sea cucumber. CSIRO, Cleveland, 70 DOI 10.13140/RG.2.1.3197.8324.

**Stout C. 2021.** Endangered species act status review report: black teatfish (*Holothuria nobilis*). Report to National Marine Fisheries Service, Office of Protected Resources, Silver Spring, MD53.

**Trianni MS, Bryan PG. 2004.** Survey and estimates of commercially viable populations of the sea cucumber *Actinopyga mauritiana* (Echinodermata: Holothuroidea), on Tinian Island, commonwealth of the Northern Mariana Islands. *Pacific Science* **58(1)**:91–98 DOI 10.1353/psc.2004.0009.

**Veronika K, Edrisinghe U, Sivashanthini K, Athauda ARSB. 2018.** Length-weight relationships of four different sea cucumber species in North-East coastal region of Sri Lanka. *Tropical Agricultural Research* **29(2)**:212–217 DOI 10.4038/tar.v29i2.8290.

**Viyakarn V, Chavanich S, Heery E, Raksasab C. 2020.** Distribution of sea cucumber, *Holothuria atra*, on the reefs in the upper Gulf of Thailand and the effect of their population densities on

sediment microalga productivity. *Estuarine, Coastal and Shelf Science* **235**:1–6 DOI 10.1016/j.ecss.2019.106514.

**Widianingsih W, Hartati R, Zainuri M, Anggoro S, Kusumaningrum HP, Mahendrajaya RT. 2019.** Morphological ossicles of sea cucumber *Paracaudina australis* from Kenjeran Waters, Surabaya, Indonesia. *IOP Conference Series: Earth and Environmental Science* **246**:1–5 DOI 10.1088/1755-1315/246/1/012019.

**Wiedemeyer WL. 1994.** Biology of small juveniles of the tropical holothurian *Actinopyga echinites*: growth, mortality, and habitat preferences. *Marine Biology* **120**:81–93 DOI 10.1007/BF00381944.

**Woo SP, Yasin Z, Ismail SH, Tan SH. 2013.** The distribution and diversity of sea cucumber in the coral reefs of the South China Sea, Sulu Sea, and Sulawesi Sea. *Deep Sea Research Part II: Topical Studies in Oceanography* **96**:13–18 DOI 10.1016/j.dsr2.2013.04.020.

**Yamada K, Sasaki K, Harada Y, Isobe R, Higuchi R. 2002.** Constituents of Holothuroidea, 12.1) Isolation and structure of glucocerebrosides from the sea cucumber *Holothuria pervicax*. *Chemical and Pharmaceutical Bulletin* **50(11)**:1467–1470 DOI 10.1248/cpb.50.1467.

**Yamana Y, Hamano T, Goshima S. 2009.** Seasonal distribution pattern of adult sea cucumber *Apostichopus japonicus* (Stichodidea) in Yoshimi Bay, western Yamaguchi Prefecture, Japan. *Fisheries Science* **75(3)**:585–591.

**Yu Z, Hu C, Zhou Y, Li H, Peng P. 2013.** Survival and growth of the sea cucumber *Holothuria leucospilota* Brandt: a comparison between suspended and bottom cultures in a subtropical fish farm during summer. *Aquaculture Research* **44**:114–124 DOI 10.1111/j.1365-2109.2011.03016.x.

**Zhong S, Qiao Y, Zhao L, Huang G, Liu Y, Huang L. 2021.** Characterization and phylogenetic analysis of the complete mitochondrial genome of *Actinopyga lecanora* (Jaegar, 1833) (Holothuriida: Holothuriidae). *Mitochondrial DNA B Resource* **6(10)**:2801–2802 DOI 10.1080/23802359.2021.1970641.

**Zulfigar Y, Kwang SY, Shau-Hwai AT, Shirayaman Y. 2008.** *Field guide to the Echinoderms (sea cucumbers and sea stars) of Malaysia*. First Edition. Kyoto, Japan: Kyoto University Press, 9–61.