# Peer review of "Bridging the gap between sustainability and profitability: unveiling the untapped potential of sea cucumber viscera"

_PeerJ, doi:10.7717/peerj.16252_

## Round 0.1 · original submission · Major Revisions

Please authors kindly address the concerns raised by the reviewers. You will see they value your work but require important additional information to strengthen the discourse.

Look forward to your revised manuscript.

Reviewer 1 ·

Basic reporting

Reviewer comment:
In this manuscript, the authors present a review work on "Bridging the gap between sustainability and profitability: unveiling the untapped potential of sea cucumber viscera". The manuscript is quite up to the mark but cannot be published in its current form. The following comments which must be addressed before the final stage:
1. Extra latest literature review (2022 - 2023) can be added in the introduction part including their proper references.
2. To my view, this manuscript is mini/commentary review. For comprehensive review, more literatrure should be considered and the discussion part should be too wide.
3. The whole manuscript should be revised to free from grammar or typo errors.
4. The document, when analyzed on Plagiarisms software, i.e., Turnitin, is showing 17% similarity index (excluding references). As per my view, it must be lower down up to 10% or so for an article. Self-Plagiarisms are also not accepted more than 2.5%.

Experimental design

Well structured.

Validity of the findings

Novel.

Annotated reviews are not available for download in order to protect the identity of reviewers who chose to remain anonymous.

Reviewer 2 ·

Basic reporting

The manuscript 86036 “Bridging the gap between sustainability and profitability: unveiling the untapped potential of sea cucumber viscera” provides new interesting information on nutritional value of internal organs of Sea cucumber and possible application for reusing sea cucumber waste.
There are merits for suitability of publication, only after considering the following:
Overall: English requires improvement. Over all context writing should be improved grammatically and need to make it readable.

Introduction:
Authors should apply their discretion to add more detail in order to provide more justification for their study being carried out and have it supplemented with more robust synthesized relevant literature to strengthen the introductory argument. The article should include sufficient introduction and background to demonstrate how the work fits into the broader field of knowledge. For instance, authors should expand upon the knowledge gap being filled, (i) how the waste made of Sea cucumber viscera could affect the environment (ii) what is the current status of the most exploited commercial species such as Holothuria scabra and Stichopus horrens.
In general, authors should stress the overexploitation of this fishery resource and the need to manage it and optimize the use of the unwanted products of sea cucumber

Sea cucumber biology/ecology
Authors should apply their discretion to add more detail about:
(i) Spatial distributions
(ii)Range of depth and type of sea bottoms where seas cucumber can be found
It would be interesting to add info about how holothurians are affected by climate change. How effects of pH changes, water temperature, salinity may influence the population abundance of holothurians, the fertilization success, the larvae metamorphosis, the development time, the physiological performances, the respiratory rate as the temperature increases and the larvae growth rate.

Socio-economic of Sea Cucumber
Authors should apply their discretion to add more detail about:
(i) Fishery of sea cucumbers could be explored by describing how sea cucumbers are caught or harvested and how this activity helps the coastal communities in the Indo-pacific region (employment etc…)
(ii)Which countries are the most sellers of sea cucumber and from which countries sea cucumber are imported into the Asian markets

Novel Source of Nutrients
Can viscera of Sea cucumbers be used as therapeutical food?

• Figures and graphs are fine
• Tables are fine

Experimental design

no comment

Validity of the findings

no comment

Reviewer 3 ·

Basic reporting

The authors of the paper Bridging the gap between sustainability and profitability: unveiling the untapped potential of sea cucumber viscera

- Discusses the importance of sea cucumbers by introducing the general biology of sea cucumbers, the nutritional content of various body parts, and the socio-economic contribution of sea cucumbers.
- Further highlights the nutritional aspects of the often-discarded viscera, and provides some future directions into the recycling of viscera, turning waste into wealth.

Experimental design

The work discusses the importance of sea cucumbers including Sea Cucumber Biology, Nutritional Value of Sea Cucumber, Socio-economic of Sea Cucumber, Nutritional Value of Sea Cucumber Viscera, and Future Directions. However, more details are required in some parts of the work to improve the quality of the findings.

- The review will be useful for scientists, academicians, biotechnologists, and environmentalists along with research scholars doing research in various sectors after some improvement. Therefore, effective management efforts required by sea cucumbers, which are currently lacking in many places could be solved.

- Introduction improvement: We suggest that authors should provide more detail relating to “The discarding of sea cucumber viscera thus results in resource waste and environmental contamination (Babji et al., 2020). Unknown to most, sea cucumber viscera have been reported to contain various nutrients such as oligosaccharides, saponins, phenols, and flavonoids ( lines 86-89)” for sustainable and waste management enhancement.

- Missing reference: “lines 281-282” 60% 0f the production cost.

- “Line 156-157”: Advice the Authors to provide some possible type of medicine for a clear understanding. (sea cucumbers are a significant source of medicine (Zhao et al., 2018).

- Table 5: We suggest that the units might present beside the species (mg/kg) only, to avoid repetition.

Methodology:
The authors employed a reliable method. However, the survey methodology “lines 99 – 106” could be presented in a simple flowchart diagram to improve the visualization of the work.

Validity of the findings

The authors concluded according to their findings, that Sea cucumber viscera are excellent sources of nutrients and bioactive compounds such as saponins, sulfated polysaccharides, amino acids, fatty acids (especially EPA and DHA), vitamins, and minerals. However, the optimal method has not been found, which required more research in the field.

---

## Round 0.2 · Minor Revisions

Please, authors, the reviewers have considered your revised work favorably. However, a very careful English Language check is required. Kindly do a very thorough English Language check on this work. Thank you

**Language Note:** The Academic Editor has identified that the English language must be improved. PeerJ can provide language editing services - please contact us at copyediting@peerj.com for pricing (be sure to provide your manuscript number and title). Alternatively, you should make your own arrangements to improve the language quality and provide details in your response letter. – PeerJ Staff

Reviewer 1 ·

Basic reporting

The authors have reviewed the manuscript as per my comments. The manuscript can be accepted in its revised form.

Experimental design

The revised manuscript is well designed.

Validity of the findings

Novelty.

Reviewer 2 ·

Basic reporting

The authors have addressed all the previously raised concerns. However, please concern with an expert for ensuring professional English

Authors added more robust synthesized relevant literature to strengthen the introductory argument

Experimental design

Methods are described with sufficient information

Validity of the findings

The manuscript has been improved in terms of contents, by contextualizing, qualifying, and justifying the research objectives

Reviewer 3 ·

Basic reporting

The article is well presentable after improvement by the authors

Experimental design

Research questions are addressed in the revised manuscript

Validity of the findings

Authors performed valid work and novel

---

## Round 0.3 · accepted · Accept

Thank you authors for your effort to further elevate the quality of the revised manuscript. It is now acceptable for publication. Thank you very much for finding PeerJ your journal of choice, and I look forward to your future scholarly contributions.
Congratulations.